# 3-Pentylcatechol, a Non-Allergenic Urushiol Derivative, Displays Anti-*Helicobacter pylori* Activity In Vivo

**DOI:** 10.3390/ph13110384

**Published:** 2020-11-13

**Authors:** Hang Yeon Jeong, Tae Ho Lee, Ju Gyeong Kim, Sueun Lee, Changjong Moon, Xuan Trong Truong, Tae-Il Jeon, Jae-Hak Moon

**Affiliations:** 1Department of Food Science and Technology, Chonnam National University, 77 Yongbongro, Gwangju 61186, Korea; wjdgkddus@naver.com (H.Y.J.); xogh9954@naver.com (T.H.L.); wnrud0610@naver.com (J.G.K.); 2Department of Veterinary Anatomy, College of Veterinary Medicine and BK21 FOUR Program, Chonnam National University, Gwangju 61186, Korea; leese@kiom.re.kr (S.L.); moonc@chonnam.ac.kr (C.M.); 3Department of Animal Science, Chonnam National University, Gwangju 61186, Korea; trongxuan.vp@gmail.com (X.T.T.); tjeon@jnu.ac.kr (T.-I.J.)

**Keywords:** urushiol, 3-pentylcatechol, 3-pentadecylcatechol, *Helicobacter pylori*, antimicrobial, triple therapy

## Abstract

We previously reported that 3-pentylcatechol (PC), a synthetic non-allergenic urushiol derivative, inhibited the growth of *Helicobacter pylori* in an in vitro assay using nutrient agar and broth. In this study, we aimed to investigate the in vivo antimicrobial activity of PC against *H. pylori* growing in the stomach mucous membrane. Four-week-old male C57BL/6 mice (n = 4) were orally inoculated with *H. pylori* Sydney Strain-1 (SS-1) for 8 weeks. Thereafter, the mice received PC (1, 5, and 15 mg/kg) and triple therapy (omeprazole, 0.7 mg/kg; metronidazole, 16.7 mg/kg; clarithromycin, 16.7 mg/kg, reference groups) once daily for 10 days. Infiltration of inflammatory cells in gastric tissue was greater in the *H. pylori*-infected group compared with the control group and lower in both the triple therapy- and PC-treated groups. In addition, upregulation of cytokine mRNA was reversed after infection, upon administration of triple therapy and PC. Interestingly, PC was more effective than triple therapy at all doses, even at 1/15th the dose of triple therapy. In addition, PC demonstrated synergism with triple therapy, even at low concentrations. The results suggest that PC may be more effective against *H. pylori* than established antibiotics.

## 1. Introduction

*Helicobacter pylori* infection is a major public health concern worldwide. This infection occurs in the gastric mucosa of more than 50% of the world’s population [1] and it is directly associated with gastrointestinal disorders, including chronic gastritis, peptic ulcer disease, mucosa-associated lymphoid tissue (MALT) lymphoma, and gastric cancer [2,3,4,5]. Gastric cancer is the second leading cause of cancer-related mortality worldwide, following only lung cancer [6]. Furthermore, *H. pylori* infection is also associated with numerous extra-gastric disorders, such as cardiovascular, neurologic, hematologic, dermatologic, head and neck, and urogynecologic diseases, as well as diabetes mellitus and metabolic syndrome [7,8]. 

The international gold-standard treatment for *H. pylori* infection is triple therapy, comprising two antibiotics (usually selected from clarithromycin, metronidazole, amoxicillin, and tetracycline) and a proton-pump inhibitor, for 7–14 days [9,10,11]. However, the success rates of these *H. pylori* eradication therapies are less than 80%, and the failure rate of *H. pylori* eradication therapy has increased, primarily due to increased antibiotic resistance [12,13,14]. Another reason for treatment failure is patient non-adherence, owing to the complexity of the treatment: it involves the repeat administration of at least three drugs over a long period [15]. In addition, these drugs are associated with several side effects, including abdominal pain, nausea, and diarrhea [16]. The high cost of *H. pylori* treatment may also be a disadvantage [15]. Therefore, there is an urgent need for the development of safe and effective therapeutic agents for *H. pylori* infections.

Lacquer tree (*Toxicodendron vernicifluum* (Stokes) F.A. Barkley, Anacardiaceae) has been used for thousands of years as a protective surface-coating material and in traditional medicine in China, Japan, and Korea [17]. It is particularly effective for treating gastrointestinal disorders, such as gastritis and gastric cancer [17]. Urushiols are a group of compounds with alkyl side chains comprising 15 or 17 carbon atoms at the C-3 position of catechol. They are the major constituents of lacquer tree sap, accounting for 60–70% of the total content [18]. In addition to their various biological activities [19,20,21,22,23], urushiols display antimicrobial activity against *H. pylori* [24]. However, urushiols can also cause serious contact dermatitis [25,26,27], which is a limitation associated with their use.

Previously, we chemically synthesized catechol-type urushiol derivatives with different alkyl side chain lengths of –C_5_H_11_, –C_10_H_21_, –C_15_H_31_, and –C_20_H_41_ at the C-3 position (Figure 1) [28]. Among these compounds, 3-decylcatechol (–C_10_H_21_) and 3-pentadecylcatechol (PDC, natural type, –C_15_H_31_) induced contact dermatitis; however, 3-pentylcatechol (PC, –C_5_H_11_) and 3-eicosylcatechol (EC, –C_20_H_41_) did not [28]. In addition, PC and EC exhibited strong antioxidative activity and high affinity for phospholipid membranes [28]. Notably, PC demonstrated enhanced antimicrobial effects in agar and broth cultures against various microorganisms involved in food spoilage and pathogenicity [29]. In addition, PC inhibited *H. pylori* to a greater extent than nalidixic acid, erythromycin, tetracycline, and ampicillin, which have been used in *H. pylori* eradication therapy [29]. Moreover, unlike PDC (Part I), PC was absorbed in the blood after oral administration [30,31]. Therefore, PC is expected to effectively eradicate *H. pylori* in gastric tissue. In this study, the in vivo antimicrobial activity of PC against *H. pylori* was evaluated and compared to that of triple therapy.

## 2. Results

### 2.1. Confirmation of Infection and Associated Gastric Disorders after H. pylori Inoculation

After 30 days of *H. pylori* Sydney Strain-1 (SS-1) inoculation, the colonization of *H. pylori* and associated gastric disorders in mouse gastric tissue were confirmed via quantitative polymerase chain reaction (qPCR) and histological analysis. The relative mRNA expression of the inflammatory cytokines, tumor necrosis factor alpha (Tnfα) and interleukin-1 beta (Il-1β) was upregulated to a greater extent in mice in the infected group than in mice in the control group (Figure 2). In addition, the expression of *H. pylori*-related genes, urease subunit alpha (ureA) and cytotoxin-associated gene A (cagA), was detected in mice in the infected group, but not in mice in the control group (Figure 2). Histological analysis revealed characteristics of gastritis, including inflammatory cell infiltration, erosion, and catarrhal inflammation in the gastric tissue of the infected group (Figure 3). These results indicate that *H. pylori* successfully colonized the stomachs of mice after inoculation and induced gastric disorders. Therefore, this animal model was used to investigate the in vivo antimicrobial activity of PC against *H. pylori*.

### 2.2. Effect of PC on the Gastric Tissue Histology of H. pylori-Infected Mice

We evaluated and graded the level of inflammatory cell infiltration in the gastric mucosa of *H. pylori*-infected mice via hematoxylin and eosin (H&E) staining (Figure 4). Grades of 0 to 3 were assigned, as follows: 0, normal; 1, mild; 2, moderate; 3, marked. All mice in the uninfected control group displayed a score of 0 (no infiltration of inflammatory cells), whereas those in the infected group displayed a score of 2 (moderate infiltration of inflammatory cells) and 3 (marked infiltration of inflammatory cells) in two mice each. The inflammation scores in all treatment groups were lower than those in the infected group. Interestingly, the scores were lower in mice treated with low doses of PC (1 and 5 mg/kg) compared with those treated with triple therapy.

### 2.3. Effect of PC on H. pylori Eradication and Cytokine Expression

To assess the effect of PC therapy on *H. pylori* eradication, the mRNA expression of the *H. pylori* markers cagA, ureA, and neutrophil-activating protein A (napA) was assessed in pyloric antrum tissue via qPCR. As shown in Figure 5, all three *H. pylori*-related transcripts were detected in the infected mice, but not in the uninfected control mice. This suggests that PC can effectively eradicate *H. pylori,* even at a dose at 1/15th of the antibiotics used in triple therapy. This response was also observed when analyzing the mRNA expression of inflammatory cytokines Tnfα and Il-1β in the pyloric antrum tissue (Figure 6). The expression of both Tnfα and Il-1β was markedly upregulated in the infected group compared with the uninfected control group; however, these genes were significantly downregulated upon PC treatment and in the reference groups. Moreover, PC treatment reduced the levels of two inflammatory cytokines more efficiently than triple therapy. Notably, in mice treated with 1 and 5 mg/kg of PC, the mRNA expression of Tnfα and Il-1β was downregulated, similar to the observation in the uninfected control mice. These results suggest that PC effectively eradicates *H. pylori* in the gastric mucosa and also helps alleviate gastrointestinal disorders at much lower concentrations than conventional antibiotics.

### 2.4. Synergistic Effect of PC in Combination with Triple Therapy

Next, we evaluated the in vivo efficacy of PC in combination with triple therapy. The expression of *H. pylori*-related genes (cagA, ureA, and napA) was not completely suppressed in the triple therapy group when the antibiotic concentration was decreased (Figure 7). In contrast, when PC was administered with triple therapy, the expression of the *H. pylori*-related genes was not observed with all concentrations (Figure 7).

Next, we evaluated the synergistic effect of PC and triple therapy on the inflammatory response (Figure 8). When mice were treated with triple therapy alone, inflammation was not completely suppressed. However, when mice were treated with PC and triple therapy, cytokine expression decreased to a level similar to that observed in the uninfected control group. These results indicate that PC demonstrated synergism with conventional antibiotic therapy, suggesting that the use of antibiotics can be reduced in the treatment of *H. pylori*.

### 2.5. Hepatotoxicity of PC

Plasma glutamate pyruvate transaminase (GPT) and glutamate oxaloacetate transaminase (GOT) levels were determined using commercial ELISA kits to evaluate the in vivo toxicity of PC after oral administration (Figure 9). No significant differences were observed between the PC-treated groups and the uninfected control group. These results indicate that PC does not cause liver toxicity.

## 3. Discussion

Urushiols are major constituents present in high concentrations in lacquer tree sap [18], with antimicrobial activity against *H. pylori* [24]. However, urushiols induce contact dermatitis [25,26,27], thereby limiting their application. 

Previously, PC, a non-allergenic urushiol derivative (Figure 1), was chemically synthesized [28], and its antimicrobial activity against various food spoilage and pathogenic microorganisms was determined [29]. PC displayed marked antimicrobial effects in both agar and broth cultures [29]. In addition, PC demonstrated greater anti-*H. pylori* activity than nalidixic acid, erythromycin, tetracycline, and ampicillin, which have been widely used to eradicate *H. pylori* [29]. In the present study, we investigated the in vivo antimicrobial activity of PC against *H. pylori* and compared it with triple therapy, which is considered the international gold-standard treatment for *H. pylori* infections.

C57BL/6 mice were inoculated with *H. pylori* SS-1 to generate a model of *H. pylori* infection. In the pyloric antrum tissue, the increased expression of Tnfα and Il-1β mRNA, which are involved in *H. pylori*-induced inflammation [32], was more prominent in the infected group than in the control group (Figure 2). In addition, characteristics of gastritis were detected in the gastric tissue of the infected mice upon H&E staining (Figure 3).

To determine the in vivo antimicrobial activity of PC, mice were inoculated with *H. pylori* SS-1 for 60 days. Subsequently, three doses (1, 5, and 15 mg/kg) of PC were administered to the *H. pylori*-infected mice once daily for 10 days. Anti-*H. pylori* activity was compared between the PC-treated groups, the positive control group, the triple therapy group, and the group receiving PDC, a natural urushiol derivative. Mortality and inflammation upon *H. pylori* infection were assessed via qPCR and histological analysis of the pyloric antrum tissue, the major habitat of *H. pylori* [33].

Histological analysis of the gastric tissues following H&E staining (Figure 4) revealed that all uninfected mice appeared normal; in contrast, inflammatory cell infiltration increased in the infected group. Inflammation scores were reduced upon PC treatment, which was more effective than triple therapy (Figure 4).

The CagA toxin, encoded by cagA, is one of the most widely studied *H. pylori* virulence factors. The CagA effector protein is injected into host target cells via a type IV secretion system and is highly associated with inflammation and the development of gastric cancer [1]. The napA encodes the NapA protein, which activates neutrophils, prevents oxidative DNA damage [34], and regulates the adhesion of *H. pylori* to stomach mucin and host epithelial cells [35]. The ureA contributes to acid resistance in *H. pylori* via the production of ammonia through the enzymatic degradation of urea in the gastric environment [1]. *H. pylori*-related genes, cagA, ureA, and napA, were analyzed via qPCR to evaluate the extent of *H. pylori* eradication (Figure 5). All three genes were detected in the infected group only and not in the uninfected groups and those receiving treatment (Figure 5). Therefore, these data indicate that *H. pylori* can be completely eradicated by PC at a much lower concentration than antibiotics. In addition, the expression of *H. pylori*-induced Tnfα and Il-1β mRNA was markedly downregulated following PC treatment (Figure 6). Moreover, the levels of these two inflammatory cytokines were effectively reduced in all the PC-treated groups compared with the triple therapy group (Figure 6).

A recent study showed that epidermal growth factor receptor signaling, implicated in gastric inflammation and carcinogenesis, remains activated following the eradication of *H. pylori* by antibiotics [36]. In addition, clarithromycin does not affect the expression of inflammatory markers in patients with atherosclerosis [37]. Knoop et al. (2016) reported that antibiotic therapy accelerates inflammation via the translocation of native intestinal bacteria [38]. Our results indicate that PC not only eradicates *H. pylori* but also improves *H. pylori*-induced gastritis. Although further studies are required to investigate the underlying mechanism of action, these results reflect the strong antioxidant activity and amphipathic structure of PC [28,39]. In addition, despite using a low concentration of PC, synergistic effects were observed with triple therapy (Figure 7 and Figure 8). Thus, PC can markedly reduce the concentration of antibiotics used and can overcome issues associated with the misuse of antibiotics [16]. In addition, the poor treatment compliance of patients owing to the need to take large amounts of antibiotics, which is a major obstacle in the antibiotic treatment of *H. pylori* infections [15], can be improved.

The plasma levels of liver transaminases, GOT and GPT, are useful biomarkers of liver injury. These enzymes are released in the blood upon hepatocyte necrosis due to acute hepatitis, ischemic injury, or toxic injury [40]. In the present study, plasma GPT and GOT levels were determined following the oral administration of PC. No evidence of liver toxicity was observed following treatment with PC (Figure 9).

## 4. Materials and Methods

### 4.1. Chemicals

3-Pentylcatechol (PC) and 3-pentadecylcatechol (PDC) were chemically synthesized in accordance with our previous method [28]. Clarithromycin, metronidazole, and omeprazole were purchased from TCI Chemical Industry (Tokyo, Japan). All other chemicals and solvents were of analytical grade, unless specified otherwise.

### 4.2. H. pylori Strain and Culture Conditions

Mouse-adapted *H. pylori* Sydney Strain-1 (SS-1) was obtained from the Korean Culture Center of Microorganisms (KCCM, Seoul, Korea) and cultured on Columbia agar or in broth medium (MB cell, Seoul, Korea), containing 5% horse serum (Gibco, Gaithersburg, MD, USA). The culture was incubated at 37 °C in a 10% CO_2_ incubator (MCO175, Sanyo, Osaka, Japan), and the bacteria were sub-cultured every 72 h [29]. Culture purity was assessed regularly.

### 4.3. Animals and Infection

All experimental procedures were approved by the Institutional Animal Care and Use Committee of Chonnam National University (no. CNU IACUC-YB-2012-26). Four-week-old C57BL/6 male mice were purchased from Samtako Bio Korea (Osan, Korea). Mice were reared in an environmentally controlled animal facility, operating on a 12:12 h dark/light cycle at 20 ± 1 °C and 55 ± 5% humidity, with ad libitum access to water and standard laboratory chow (Harlan Rodent diet, 2018S, by Samtako Bio Korea) [24]. 

Four mice per group were inoculated with *H. pylori* SS-1, which can effectively colonize the mouse gastric mucosa [41]. A total of 100 µL aliquots (10^8^ CFU) of Columbia broth were administered to the mice for 60 days, three times every 2 days, using a zonde needle. After 30 days of inoculation, three mice were sacrificed to confirm infection. Blood was withdrawn from the abdominal aorta of the mice under light anesthesia (isoflurane) and collected in heparinized tubes. Plasma was obtained via centrifugation (2767× *g*, 4 °C, 15 min). The pyloric antrum of the stomach was harvested for quantitative polymerase chain reaction (qPCR) and histological analysis. Uninfected mice were administered the same volume of fresh Columbia broth; this group was considered the negative control. All samples were stored at −80 °C until use.

### 4.4. PC Treatment after H. pylori Infection

Following 60-day *H. pylori* inoculation, PC was administered to the infected mice with 100 μL of water once daily for 10 days [24]. Triple therapy and PDC, a natural form of urushiol, were used as reference groups. Infected mice were divided into seven experimental groups (*n* = 4): control group (uninfected, negative control); *H. pylori*-infected group; *H. pylori* infection + triple therapy treatment group; *H. pylori* infection + PDC 26.7 mg (83.3 μmol)/kg treatment group; *H. pylori* infection + PC 1 mg (5.6 μmol)/kg treatment group; *H. pylori* infection + PC 5 mg (27.8 μmol)/kg treatment group; and *H. pylori* infection + PC 15 mg (83.3 μmol)/kg treatment group. Triple therapy comprised omeprazole (700 μg/kg), metronidazole (16.7 mg/kg), and clarithromycin (16.7 mg/kg). After 10 days of treatment, the mice were euthanized and samples were harvested as described above.

To confirm the synergistic effect of PC with triple therapy, triple therapy was administered at four concentrations, as follows: existing concentration (metronidazole and clarithromycin: 16.7 mg/kg; omeprazole: 700 μg/kg), one-fifth, one-tenth, and one-fifteenth of the existing concentration. In contrast, PC was administered at the same concentration (1 mg/kg). In accordance with the above conditions, triple therapy and PC were administrated orally to the *H. pylori*-infected mice once daily for 5 days. The control and infection groups received distilled water under the same conditions. After 5 days of treatment, the mice were euthanized and samples were harvested, as described above.

### 4.5. Histological Examination

Gastric tissue was fixed in 4% (*w*/*v*) paraformaldehyde (PFA) in phosphate-buffered saline (PBS, pH 7.4) for 24 h, dehydrated in a graded ethanol series (70%, 80%, 90%, 95%, and 100%), cleared in xylene, embedded in paraffin, and sectioned into 5-μm-thick slices. Serial sections were stained with hematoxylin and eosin (H&E) and examined microscopically to determine whether the gastric mucosa contained any pathological lesions [42].

### 4.6. RNA Analysis

Total RNA was isolated from mouse gastric tissue using the TRI Reagent^®^ (Molecular Research Center, Cincinnati, OH, USA). cDNA was synthesized using the ReverTra Ace^®^ qPCR RT kit (Toyobo, Osaka, Japan), and qPCR amplification was accomplished using a Mx3000P qPCR System (Agilent Technologies, Santa Clara, CA, USA). Primer sequences are listed in Table 1. mRNA expression levels were normalized to those of the mouse ribosomal protein, Large, P0 (Rplp0), as the internal control, determined via the comparative threshold cycle method [43].

### 4.7. Determination of Plasma Glutamate Oxaloacetate Transaminase (GOT) and Glutamate Pyruvate Transaminase (GPT) Levels

Plasma GPT and GOT levels were determined using GPT and GOT enzyme-linked immunosorbent assay (ELISA) kits (Asan Pharmaceutical, Seoul, Korea) in accordance with the manufacturer’s instructions.

### 4.8. Statistical Analysis

Data are presented as the mean ± standard deviation and were determined using Statistical Package for Social Sciences (SPSS, IBM, Armonk, NY, USA) version 19.0. Statistically significant differences were ascertained using one-way analysis of variance, followed by the Tukey–Kramer and Student’s *t*-tests. *p* < 0.05 was considered significant.

## 5. Conclusions

In summary, we compared the in vivo antimicrobial effects of PC and conventional triple therapy against *H. pylori* using a mouse model of *H. pylori* infection. PC completely eradicated *H. pylori*, even when administered at a dose 1/15th that of conventional antibiotics used for triple therapy. In addition, gastritis was rapidly alleviated upon PC treatment. Thus, PC may be a potential viable alternative to triple therapy for *H. pylori* and gastrointestinal disorders.

## Figures and Tables

**Figure 1 pharmaceuticals-13-00384-f001:**
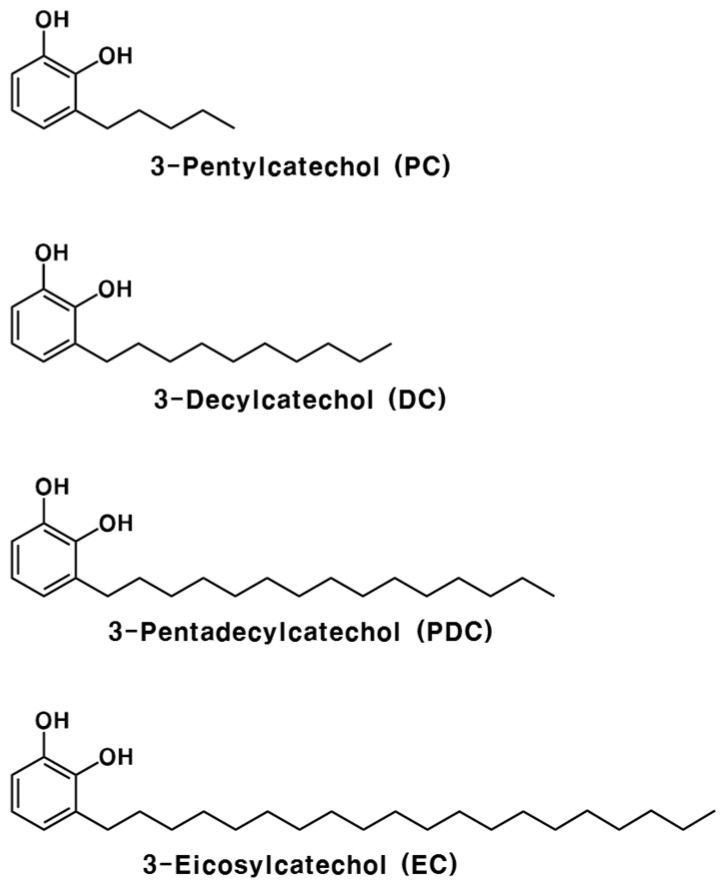
Structures of synthesized urushiol derivatives.

**Figure 2 pharmaceuticals-13-00384-f002:**
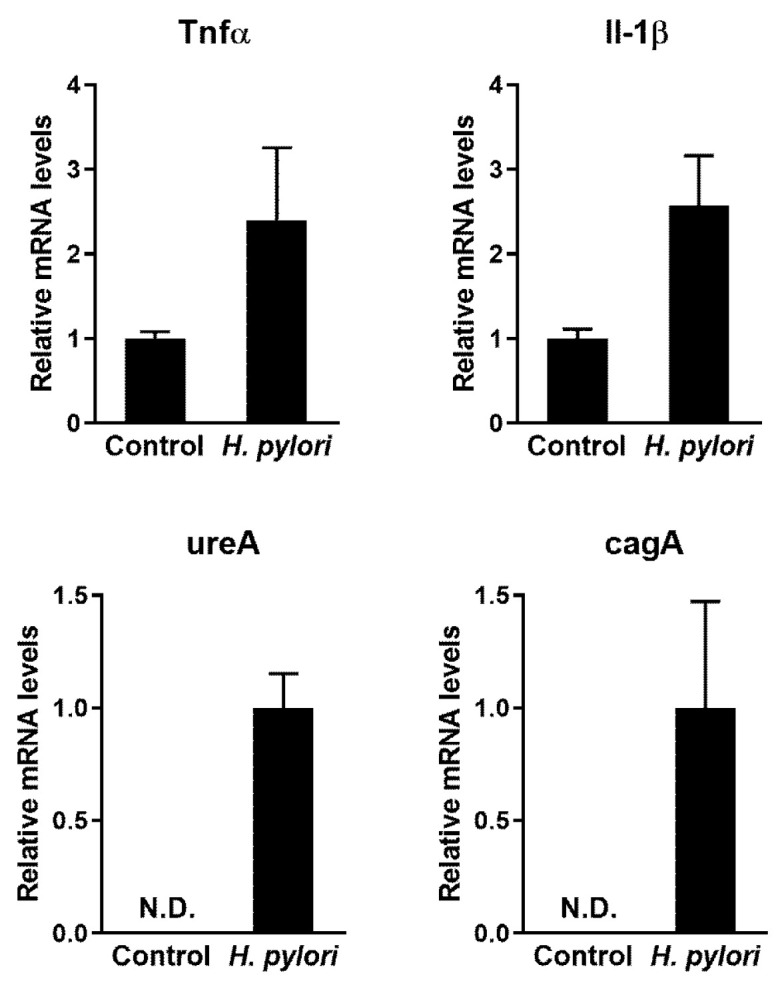
Expression levels of tumor necrosis factor alpha (Tnfα), interleukin-1 beta (Il-1β), urease subunit alpha (ureA), and cytotoxin-associated gene A (cagA) mRNA in mouse gastric tissue following *Helicobacter pylori* Sydney Strain-1 (SS-1) inoculation. *H. pylori* SS-1 was administered to C57BL/6 mice for 30 days. N.D., not detected.

**Figure 3 pharmaceuticals-13-00384-f003:**
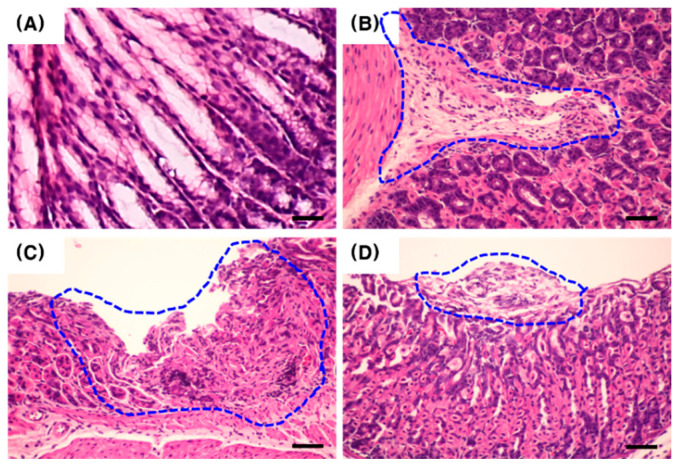
Histological analysis of hematoxylin and eosin-stained mouse gastric tissue after *H. pylori* SS-1 inoculation. *H. pylori* SS-1 was administered to C57BL/6 mice for 30 days. (**A**) uninfected control; (**B**) inflammatory cell infiltration (dotted line) in an *H. pylori*-infected mouse; (**C**) erosion (dotted line) in an *H. pylori*-infected mouse; (**D**) catarrhal inflammation (dotted line) in an *H. pylori*-infected mouse. Scale bar = 20 μm.

**Figure 4 pharmaceuticals-13-00384-f004:**
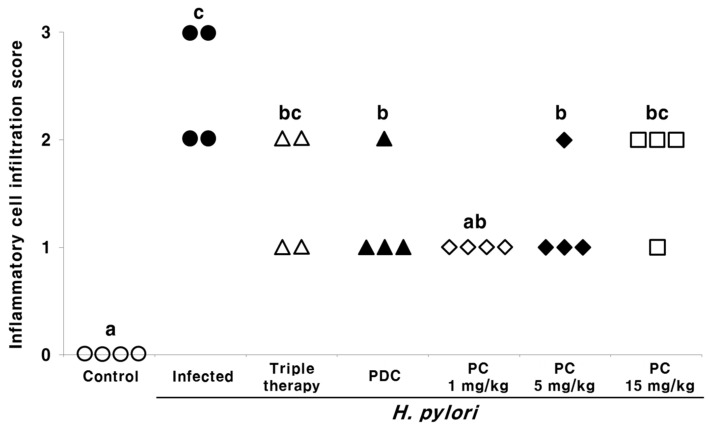
Effect of 3-pentylcatechol (PC) treatment on the histology of gastric tissue from *H. pylori*-infected mice (hematoxylin and eosin staining). Inflammatory cell infiltration was graded from 0 to 3: 0, normal; 1, mild; 2, moderate; 3, marked. ○, inflammatory cell infiltration score of control group; ●, inflammatory cell infiltration score of *H. pylori*-infected group; Δ, inflammatory cell infiltration score of *H. pylori* + triple therapy-treated group; ▲, inflammatory cell infiltration score of *H. pylori* + 3-pentadecylcatechol (PDC)-treated group; ◊, inflammatory cell infiltration score of *H. pylori* + 1 mg/kg of PC-treated group; ♦, inflammatory cell infiltration score of *H. pylori* + 5 mg/kg of PC-treated group; □, inflammatory cell infiltration score of *H. pylori* + 15 mg/kg of PC-treated group. Different letters (a, b, and c) indicate a significant difference (*p* < 0.05), ascertained via the Tukey–Kramer test.

**Figure 5 pharmaceuticals-13-00384-f005:**
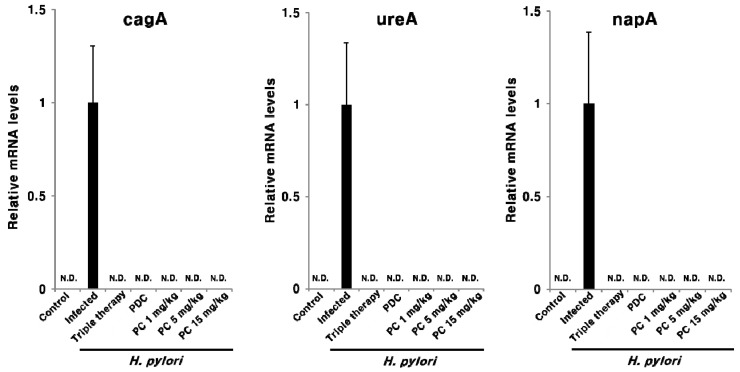
Effect of PC treatment on the expression of *H. pylori* cagA, ureA, and napA in the gastric tissue of the *H. pylori*-infected mice. N.D., not detected.

**Figure 6 pharmaceuticals-13-00384-f006:**
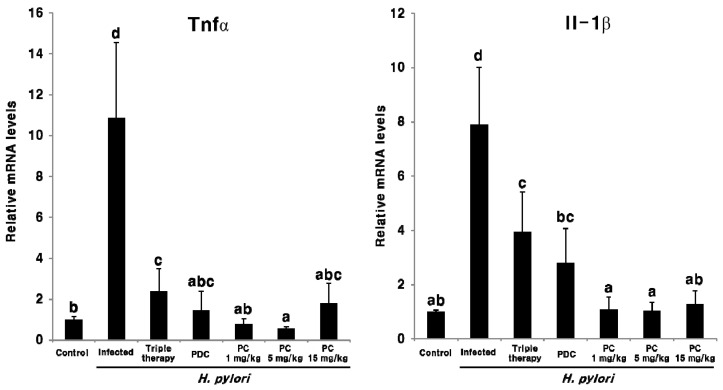
Effect of PC treatment on the expression of Tnfα and Il-1β mRNA in the gastric tissue of the *H. pylori*-infected mice. Different letters (a, b, c, and d) indicate a significant difference (*p* < 0.05), ascertained via the Tukey–Kramer test. PC, 3-pentylcatechol; PDC, 3-pentadecylcatechol.

**Figure 7 pharmaceuticals-13-00384-f007:**
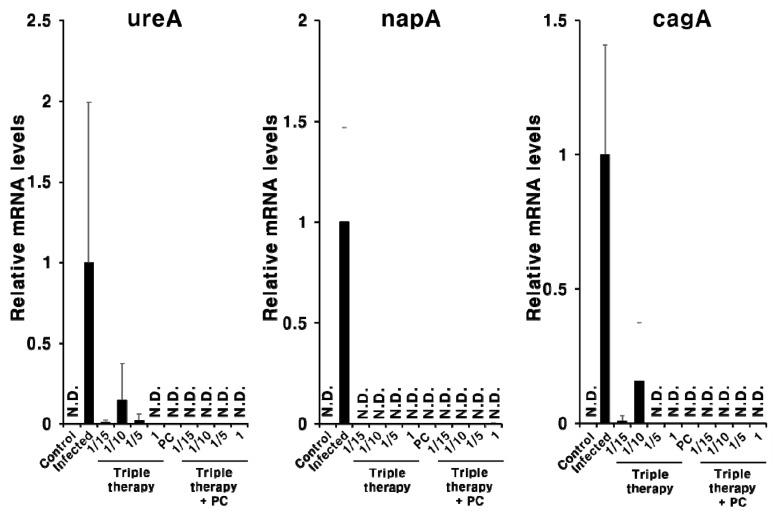
Expression of *H. pylori* ureA, napA, and cagA mRNA in the gastric tissue of the *H. pylori*-infected mice following combination treatment with PC and triple therapy. Different letters indicate a significant difference (*p* < 0.05), ascertained via the Tukey–Kramer test. Triple therapy was administered at four concentrations. 1, Existing concentration of triple therapy (metronidazole and clarithromycin: 16.7 mg/kg; omeprazole: 700 μg/kg); 1/5, one-fifth of the existing concentration of triple therapy; 1/10, one-tenth of the existing concentration of triple therapy; 1/15, one-fifteenth of the existing concentration of triple therapy. PC was administered at 1 mg/kg. N.D., not detected.

**Figure 8 pharmaceuticals-13-00384-f008:**
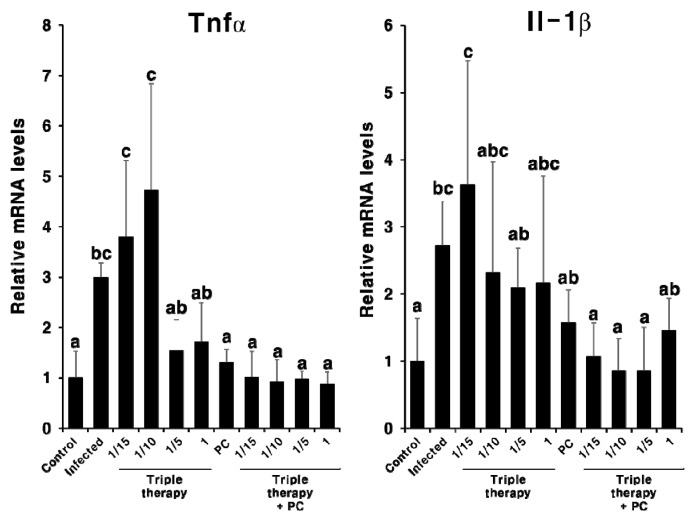
Expression levels of Tnfα and Il-1β mRNA in the gastric tissue of the *H. pylori*-infected mice after combination treatment with PC and triple therapy. Different letters (a, b, and c) indicate a significant difference (*p* < 0.05), ascertained via the Tukey–Kramer test. Triple therapy was administered at four concentrations. 1, Existing concentration of triple therapy (metronidazole and clarithromycin: 16.7 mg/kg; omeprazole: 700 μg/kg); 1/5, one-fifth of the existing concentration of triple therapy; 1/10, one-tenth of the existing concentration of triple therapy; 1/15, one-fifteenth of the existing concentration of triple therapy. PC was administered at 1 mg/kg.

**Figure 9 pharmaceuticals-13-00384-f009:**
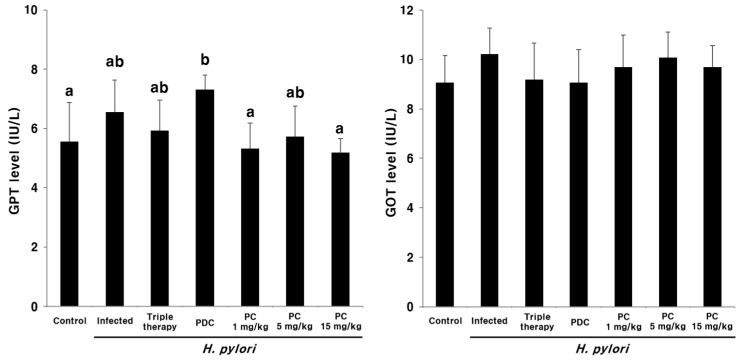
Plasma glutamate pyruvate transaminase (GPT) and glutamate oxaloacetate transaminase (GOT) levels after 3-pentylcatechol treatment. Different letters (a and b) indicate a significant difference (*p* < 0.05), ascertained via the Tukey–Kramer test.

**Table 1 pharmaceuticals-13-00384-t001:** Primers used in this study.

Gene	Sequence	
	Forward	Reverse
Rplp0	GTGCTGATGGGCAAGAAC	AGGTCCTCCTTGGTGAAC
Tnfα	CGAGTGACAAGCCTGTAGCC	AGCTGCTCCTCCACTTGGT
Il-1β	ATGAGAGCATCCAGCTTCAA	TGAAGGAAAAGAAGGTGCTC
cagA	CCGATCGATCCGAAATTTTA	CGTTCGGATTTGATTCCCTA
ureA	TGTTGGCGACAGACCGGTTCAAATC	GCTGTCCCGCTCGCAATGTCTAAGC
napA	CCATGTGCATAAAGCCACTG	GAGTTTGAGCGCTTCGGATA

Ribosomal protein (Rplp0), Large, P0; tumor necrosis factor alpha (Tnfα); interleukin-1 beta (Il-1β); cytotoxin-associated gene A (cagA); urease subunit alpha (ureA); neutrophil-activating protein A (napA).

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
