# Peer review of "3-Pentylcatechol, a Non-Allergenic Urushiol Derivative, Displays Anti-Helicobacter pylori Activity In Vivo"

_pharmaceuticals, 2020, doi:10.3390/ph13110384_

Round 1

Reviewer 1 Report

The manuscript investigated the antimicrobial activity of 3-Pentylcatechol against Helicobacter pylori in mice based on previous in vitro study. It is fairly well designed and the results are supportive, however, there are still several questions need to be addressed.

1. Figure 5 and 6, the PC groups shows similar level with ctrl group for all the biomarkers, however, in figure 4, the score of PC groups was higher than control group, could the authors explain this?

2. Could the author explain the reason of the initial dose level selection for PC group for the in vitro test?

3. 1 mg/kg of PC already showed very significant effect for improving all the biomarkers measured in prior steps, so maybe a lower dose is a better choice for the synergistic effect assay.

4. In figure 2, for UerA and CagA, the value of the control groups is N.D., what is the reference the authors used to make the relative Y-axis?

Author Response

<Reviewer 1>

The manuscript investigated the antimicrobial activity of 3-Pentylcatechol against Helicobacter pylori in mice based on previous in vitro study. It is fairly well designed and the results are supportive, however, there are still several questions need to be addressed.

Figure 5 and 6, the PC groups shows similar level with ctrl group for all the biomarkers, however, in figure 4, the score of PC groups was higher than control group, could the authors explain this?

<Answer> Figure 4 shows the degree of infiltration of phagocytic cells into tissues as scores. In addition, the increase in macrophage infiltration caused by Helicobacter infection is well known [1,2]. Therefore, although the Helicobacter infection was completely cured, it is suggested that M2 macrophage may be still remained because it is recovering from chronic inflammation [3].

[1] Kaparakis M, et al. Macrophages are mediators of gastritis in acute Helicobacter pylori infection in C57BL/6 mice. Infect Immun. 2008;76(5):2235-2239.

[2] Whitney AE, et al. Increased macrophage infiltration of gastric mucosa in Helicobacter pylori-infected children. Dig Dis Sci. 2000;45(7):1337-1342.

[3] Wang S, et al. Antibiotics induce polarization of pleural macrophages to M2-like phenotype in patients with tuberculous pleuritis. Sci Rep. 2017;7(1):14982.

Could the author explain the reason of the initial dose level selection for PC group for the in vitro test?

<Answer> The present study is no in vitro test. Therefore, we understood “in vitro” in the reviewer’s comment as an in vivo test. First, we established the administration concentration of antibiotics. That is, two antibiotics prescribing for the treatment of H. pylori are 1 g each/60 kg human body weight. It is corresponding to 16.7 mg/kg. Therefore, the treatment concentration was established to 15 mg with reflection of molecular weight. In addition, since two antibiotics are used for triple therapy, the treatment weight of two antibiotics can be calculated by combining them. However, the reaction mechanism of the two antibiotics is different. Therefore, we treated on the basis of one (wt.) of two antibiotics.

1 mg/kg of PC already showed very significant effect for improving all the biomarkers measured in prior steps, so maybe a lower dose is a better choice for the synergistic effect assay.

<Answer> We appreciate reviewer’s meaningful comment. In general, for the treatment of H. pylori, antibiotics should be taken for 10 days. However, in the present study, in order to confirm the synergistic effect of PC and triple therapy, we administered the drugs orally for only 5 days. As a result, PC showed sufficient effect by individual treatment of 1 mg PC. However, it showed better effect when PC was treated with the triple therapy containing gastric acid inhibitors. Therefore, it is suggested that the applicability of PC is sufficient to reduce the existing antibiotic dose.

In figure 2, for UerA and CagA, the value of the control groups is N.D., what is the reference the authors used to make the relative Y-axis?

<Answer> We appreciate very much for your valuable comments. We replaced figure 2 with graphs in which UreA and CagA values were converted based on the H. pylori-infected group.

Reviewer 2 Report

The article entitled “3-Pentylcatechol, A Non-Allergenic Urushiol Derivative, Displays Anti-Helicobacter pylori Activity In Vivo” describes the biological evaluation of 3-alkylcatechol derivatives in mice model. The presented data confirm statistically significant antibacterial and anti-inflammatory activities of reported 3-pentylcatechol tested by histological examination, as well as RNA analysis.  Moreover, no hepatoxicity and synergistic effects with triple therapy (omeprazole, 0.7 mg/kg; metronidazole, 16.7 mg/kg; clarithromycin) were observed. The paper has correct form with logical sections and the obtained results are explained comprehensively. Although the topic is interesting and the manuscript fits within the scope of the Journal, I would like to make some considerations regarding the work:

  1. “positive control group” is not defined properly (eg. lines: 20, 106/107). In my opinion, positive control should be the infected group without treatment rather than “Triple therapy” or “PDC”. These latter two should be described as “reference groups”.
  2. Figure 4 suggests that the lower concentration of PC is, the better effect produces. Could authors comment on dose-independent behavior? Similar situation refers to triple therapy in Figure 7.
  3. In the experiment mRNA fragments levels were determined. Line 194: please, replace the word “genes” with “transcripts”.

Summing up, I strongly recommend the paper to be published in Pharmaceuticals after some minor revisions.

Author Response

<Reviewer 2>

The article entitled “3-Pentylcatechol, A Non-Allergenic Urushiol Derivative, Displays Anti-Helicobacter pylori Activity In Vivo” describes the biological evaluation of 3-alkylcatechol derivatives in mice model. The presented data confirm statistically significant antibacterial and anti-inflammatory activities of reported 3-pentylcatechol tested by histological examination, as well as RNA analysis. Moreover, no hepatoxicity and synergistic effects with triple therapy (omeprazole, 0.7 mg/kg; metronidazole, 16.7 mg/kg; clarithromycin) were observed. The paper has correct form with logical sections and the obtained results are explained comprehensively. Although the topic is interesting and the manuscript fits within the scope of the Journal, I would like to make some considerations regarding the work:

“positive control group” is not defined properly (eg. lines: 20, 106/107). In my opinion, positive control should be the infected group without treatment rather than “Triple therapy” or “PDC”. These latter two should be described as “reference groups”.

<Answer> We revised according to the reviewer’s comment.

Figure 4 suggests that the lower concentration of PC is, the better effect produces. Could authors comment on dose-independent behavior? Similar situation refers to triple therapy in Figure 7.

<Answer> PC possesses very strong reducing power and antioxidant activity. Therefore, the high concentration may affected the wound area caused by the infection. It is also considered that it may be due to excessive secretion of gastric acid. Therefore, it is suggested that it may show a better effect if prescribed with a gastric acid suppressant through further study is needed. The debate over the dosage levels of drugs is continued [1,2]. This result, which showed high efficacy at low concentrations, is considered more positive as it can reduce overdose and side effects of drugs.

[1] Dimmitt SB and Stampfer HG. Low drug doses may improve outcomes in chronic disease. Med J Aust. 2009;191(9):511-513.

[2] McCormack JP, et al. Is bigger better? An argument for very low starting doses. CMAJ. 2011;183(1):65-69.

In the experiment mRNA fragments levels were determined. Line 194: please, replace the word “genes” with “transcripts”.

<Answer> We revised according to the reviewer’s comment.

Summing up, I strongly recommend the paper to be published in Pharmaceuticals after some minor revisions.

<Answer> We appreciate very much for your favorable comments.